



# Airborne Gravimetry with Quantum Technology: Observations from Iceland and Greenland

Tim Enzlberger Jensen[1], Bjørnar Dale[1], Andreas Stokholm[1], René Forsberg[1], Alexandre Bresson[2], Nassim Zahzam[2], Alexis Bonnin[2], and Yannick Bidel[2]

[1]Department of Space Research and Technology, Technical University of Denmark (DTU Space), Elektrovej 328, 2800 Kgs. Lyngby, Denmark
[2]DPHY, ONERA, Université Paris Saclay, Palaiseau, France

**Correspondence:** Tim Enzlberger Jensen (timj@space.dtu.dk)

**Abstract.** We report on the availability of data from an airborne gravity campaign in Iceland and Greenland, conducted during June and July 2023. The dataset includes observations from a platform stabilised gravimeter based on cold-atom quantum technology and a strapdown gravimeter based on classical technology. The data is available in three different levels of processing making it relevant to users interesting in working with "quantum" and "hybrid" data as well as users interested in geophysical

5  studies. The mansucript describes the data processing applied to derive the various levels of data and presents an evaluation of the data accuracy. This evaluation indicates an accuracy of 1 to 2 mGal for both sensors, depending on the roughness of the gravity field. Although the two technologies lead to similar performance, further analysis indicates that the error characteristics are different and that final estimates would benefit from a combination.

## 10  1  Introduction

This manuscript reports on the data available from and airborne gravity campaign carried out in summer 2023. Airborne gravimetry is a powerful tool for measuring gravity at regional scales where ground measurements are difficult, such as mountains or coastal areas. Recently, a new technology of airborne gravimeter based on cold-atom quantum technology has emerged. Unlike classical technologies that can only measure the variation of gravity from an aircraft, a quantum gravimeter provides

15  a direct absolute measurement of gravity, eliminating the need for calibration and drift estimation. Here we present airborne gravity measurements over Iceland and Greenland using a quantum and a classical gravimeter.

The campaign was a result of merging two projects targeting locations in Iceland and Greenland, but executed using the same instrumentation. The main instrument is the GIRAFE quantum gravimeter developed by the French Aerospace Lab (ONERA), appended by the iMAR iNAT-RQH strapdown gravimeter (Jensen (2024)) owned by the Department of Space Research and

20  Technology at the Technical University of Denmark (DTU Space). Two similar campaigns have been carried out previously in 2017 (Iceland) and 2019 (France), both of which remain the only of their kind reported in the literature (Bidel et al. (2020),Bidel



et al. (2023)). The current campaign contains a repetition of the previous campaign in Iceland, making it possible to directly evaluate the improvements implemented since the first airborne test in 2017.

With the GIRAFE sensor being the only of its kind, the technology remains at a development stage. For this reason, the data is made available at three different levels of processing, making it relevant for several users. The level 0 data is the raw observations collected during the campaign, relevant for users aiming to perform their own data processing. The level 1 data contains several intermediate variables, relevant for users aiming to develop their own processing software or interested in tuning some processing parameters for specific purposes. The level 2 data contains the final gravity estimates from the campaign, relevant for users aiming at geophysical studies or gravity field modelling. The data is made available by the European Space Agency (ESA) and these data levels are similar to other data products available through their servers (Jensen et al. (2024)).

Following the introduction, section 2 gives a brief description of the airborne campaign and the instrumentation. Section 3 introduces some fundamental theory and data processing strategy applied to transition from level 0 til level 2 data. Additionally, some means of evaluating the accuracy of the level 2 gravity estimates is described. Finally, the obtained results are presented in section 4, which also aims to give some insights into the characteristics of the data. This section is followed by a conclusion and a statement of data availability.

## 2 Survey Overview and Instrumentation

As a result of merging two projects, the measurement campaign consists of two separate geographical locations, namely Iceland and Greenland. The base airport of operation was located in Akureyri (AEY) in Iceland from where all instruments were installed in the chartered DHC-6 300 Twin-Otter aircraft. Following installation and testing, survey flights in Iceland were carried out. These flight are part of the AirQuantumGrav project, funded by ESA. Afterwards, the aircraft was transfered to Nuuk in Greenland, from where a survey of the surrounding fjord system was carried out. This second part of the campaign belongs to the Green Quantum project, funded by the Danish Ministry of Defense Acquisition and Logistics Organization (DALO). Below is a short overview of the two projects along with the instrumentation installed onboard the aircraft.

### 2.1 The AirQuantumGrav Project

The main objective of the AirQuantumGrav project was to pursue the quantum technological benefits of stability, absolute measurements and no calibration needs for airborne gravimetry. The project builds upon an earlier project such that flight lines from a previous campaign in 2017 are repeated (Bidel et al. (2020), Bresson et al. (2023)). The current project identified the following targets for the 2024 campaign:

- A repetition of the 2017 airborne campaign over the Vatnajökull ice cap

- An overflight of the Askja volcano

- An overflight of the Fagradallsfjall volcano

– A repetition of the 2017 flight line from Snæfellsjökull to Akureyri

All of the target locations are sources of mass variation due to known geophysical phenomena, i.e. melting ice cap, rapid uplift and volcanic activity. The repetition of such targets would enable the investigation of mass variation, assuming the accuracy is sufficient. An overview of the flights carried out within the project is shown in Figure 1.

**Figure 1.** Ground track from flights in Iceland. Legend refers to the day of year. Also shown is the location of the Askja and Fagradalsfjall volcanoes along with the airports of Akureyri (AEY) and Reykjavik domestic (RKV).

## 2.2 The Green Quantum Project

The Green Quantum campaign was planned as a regular grid survey, with the purpose of mapping the gravity field with a uniform spatial resolution over the area. The data collected in the area will be used as input for the computation of a new Geoid





model for the Nuuk region and later for updating the vertical reference frame of Greenland. The ground tracks from the flights

are shown in Figure 2.

**Figure 2.** Ground track from flights covering the Nuuk fjord system in Greenland. Legend refers to the day of year. Also shown is the location of the Nuuk aiport (GOH).

The campaign additionally serves to support the ADEQUADE project funded be the EU-EDF programme. For this the east-west flight line was repeated four times, testing the quantum gravimeter with the stabilising platform active and inactive.

### 2.3    Instrumentation

Onboard the aircraft was installed two gravimeters along with three different Global Navigation Satellite System (GNSS)

receivers. The main instrumentation was the GIRAFE cold-atom quantum gravimeter developed by ONERA.

The GIRAFE gravimeter is based on the measurement of the acceleration of a cloud of cold atoms in free fall using matter wave interferometry. A detailed description of this technology can be found in Tino and Kasevich (2014). The description of

GIRAFE is given in Bidel et al. (2018, 2020, 2023). The main advantage of the quantum gravimeter compared to classical technologies is the ability to provide an absolute value of the acceleration, which means that the instrument does not require calibration or drift estimation.

To represent the classical technology, an iMAR iNAT-RQH strapdown gravimeter (Jensen (2024)) was installed adjacent to the platform-based quantum gravimeter. Both sensors are illustrated in Figure 3. The strapdown gravimeter is essentially a navigation-grade Inertial Measurement Unit (IMU) that is rigidly fixed to the chassis of the aircraft. Since no active platform ensures that the sensor remains aligned with the direction of the gravity vector, the orientation of three perpendicular sensors is computed post-processing in order to resolve the gravity acceleration. This will be elaborated on in Section 3.1.

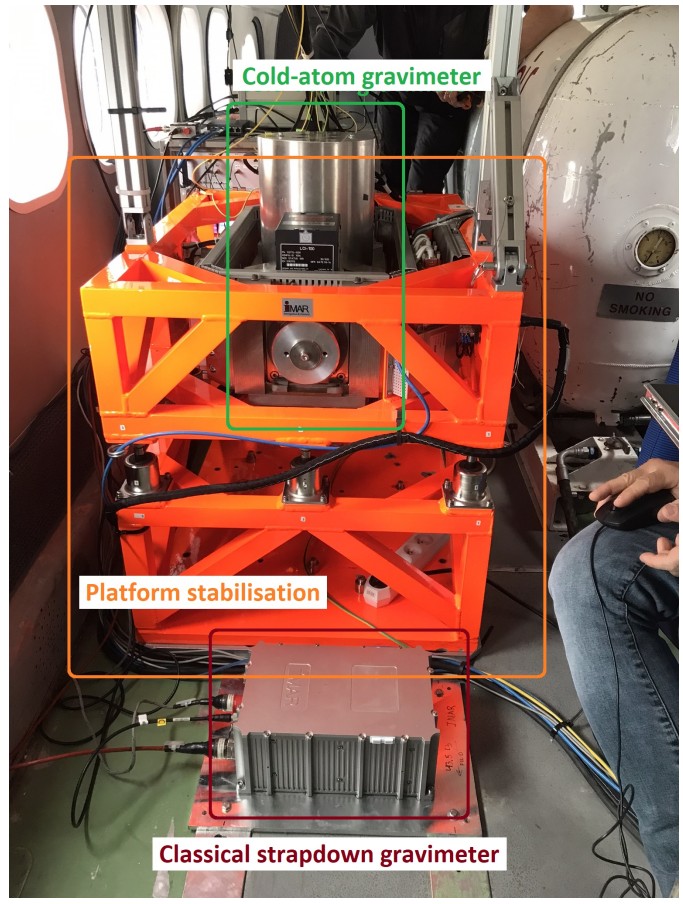

**Figure 3.** Photograph of the installation of the two gravimeters. The GIRAFE cold atom gravimeter is mounted on a stabilizing platform while the iMAR strapdown gravimeter is rigidly attached to the aircraft floor.

Additional to the gravimeters, three GNSS receivers were installed and connected to the same GNSS antenna on top of the aircraft. Contrary to stationary gravimetry, the GNSS observing system is essential to moving-base gravimetry and gravity



cannot be derived without it (or other systems to determine motion-induced accelerations). The three receivers installed in the aircraft were:

– Javad Delta GNSS receiver (GPS and GLONASS only)

      – Septentrio Mosaic-Go GNSS receiver (multi-constellation)

      – NovAtel PwrPak7 GNSS receiver (multi-constellation)

Additionally, a Javad Triumph GNSS receiver was placed in the airport whenever possible. This may serve as a base station for differential processing on most flights. Notice however that the receivers differ in which satellite constellations they are

able to receive data from and which observation type is logged on the unit.

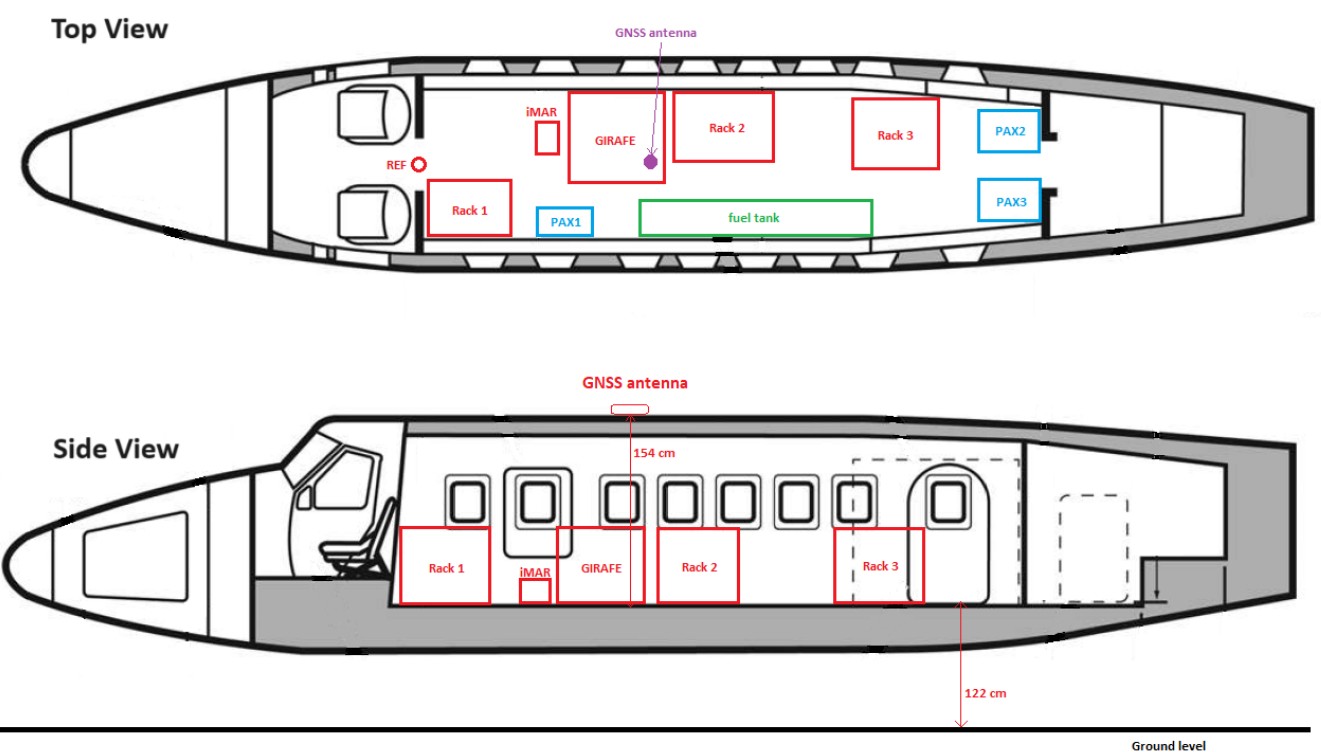

**Figure 4.** Sketch of aircraft installation with placement of iMAR and GIRAFE gravimeters relative to the GNSS antenna.

Finally, in order to power and operate the various equipment, several cabinets with electronics were installed on the aircraft along with the equipment. A sketch of the installation is illustrated in Figure 4.

One important parameter for the data processing is the physical separation between the gravity sensors and the GNSS antenna located on top of the aircraft. These distances, generally denoted as the lever-arm, have been estimated during data processing

and are listed in Table 1.



**Table 1.** Estimates of lever arm from the two gravity sensors to the location of the GNSS antenna. Distances are along the forward, starboard and through-the-floor directions of the aircraft.

| Gravimeter | Forward | Starboard | Through-the-floor |
|---|---|---|---|
| GIRAFE | -0.64 m | -0.76 m | -1.12 m |
| iMAR iNAT-RQH | -1.24 m | -0.64 m | -1.60 m |

## 3 Methodology

This section is initialised with a short summary of the theoretical principles of airborne gravimetry. This is followed by a section outlining the processing methodology along with two sections describing methods for internal and external validation of results. Finally, a section describing a simple method of combining estimates from two sensors is included.

### 3.1 Moving-Base Gravimetry

As mentioned in section 2.3, the essential instrumentation for the campaign are the two gravimeters. Such gravimeters measure specific force accelerations, $\mathbf{f}^s$, along the axes of a sensor reference frame ($s$-frame). The specific force represents acceleration originating from both movement and gravitation, such that

$$\mathbf{f}^i = \mathbf{C}_s^i \mathbf{f}^s = \ddot{\mathbf{r}}^i - \mathbf{g}^i \, , \tag{1}$$

where the kinematic acceleration, $\ddot{\mathbf{r}}$, is expressed as the double derivative of position, $\mathbf{r}$, with respect to time. The transformation matrix, $\mathbf{C}_s^i$, represents a rotation of the sensor-frame axis ($s$-frame) onto the axes of an inertial reference frame ($i$-frame). The above expression is valid in an inertial reference frame only, which is denoted using the superscript $i$. An accelerometer measures along a single axis in space such that three perpendicular accelerometers are needed in general to measure the full vector quantity. These three values are thus obtained along the axes of the $s$-frame, which may have any orientation in space.

Data processing is done in the navigation frame ($n$-frame), centered at the instrument location with axes along the local north, east and down directions. For this we need to first project the observed accelerations onto the axes of the $n$-frame as

$$\mathbf{f}^n = \mathbf{C}_s^n \mathbf{f}^s \, , \tag{2}$$

where $\mathbf{C}_s^n$ is a rotation operator. In practice, $\mathbf{C}_s^n$ need not be computed post-processing, but can be applied real-time by continuously rotating the instrument such that its axes are always aligned with the north, east and down directions. The stabilizing platform of the GIRAFE instrument represents such a real-time solution, although it only rotates about two axes in order to align the gravity sensor with the local vertical. The iMAR strapdown instrument on the other hand additionally contains three perpendicular gyroscopes, measuring rotations, such that the orientation of the instrument (i.e. the $\mathbf{C}_s^n$ matrix) can be computed post-mission.



Additionally, we need to modify (1) in order to comply with the $n$-frame by considering relative movement between the $i$- and $n$-frames. Here two terms arise, one originating from the rotation of the Earth with respect to inertial space and one from the rotation of the navigation frame in order to keep it aligned with the local north and down directions as we move across the surface of the Earth (Groves (2013)). Collectively, these terms are known as the Eötvös effect

$$\mathbf{E}^n = (2\boldsymbol{\Omega}_{ie}^n + \boldsymbol{\Omega}_{en}^n)\,\mathbf{v}^n \,, \tag{3}$$

where $\boldsymbol{\Omega}_{ie}^n$ and $\boldsymbol{\Omega}_{en}^n$ are skew-symmetric matrices of the rotational rates mentioned above, while $\mathbf{v}^n$ is the velocity resolved about the $n$-frame axes. As a kind of mental argument we may imagine this as a correction term due to the choice of reference frame. For example, imagine a sensor stationary on the surface of the Earth. In this case we would want the velocity to be zero and the position to be constant. However, the accelerometer measures with respect to inertial space, meaning that it will measure some acceleration due to the rotating Earth. If one would simply integrate the accelerations with respect to time, we would obtain a changing velocity and position in time. In order to cancel this effect, we need to introduce a fictitious correction term on the accelerations before integrating. This is the Eötvös effect.

Finally, we may piece together the above terms to form the following fundamental equation of gravimetry (Jekeli (2001))

$$\mathbf{g}^n = \ddot{\mathbf{r}}^n - \mathbf{C}_s^n \mathbf{f}^s + (2\boldsymbol{\Omega}_{ie}^n + \boldsymbol{\Omega}_{en}^n)\,\mathbf{v}^n \,, \tag{4}$$

noticing that the gravimeter measures, $\mathbf{f}^s$, while the position, $\mathbf{r}^n$, is obtained from GNSS observations. In this sense, GNSS becomes an integral component of the instrumentation, without which the gravity acceleration cannot be derived. Finally, as is customary in geodesy, the normal gravity vector, $\boldsymbol{\gamma}$, representing the height dependent gravity field of a simplified Earth model, is subtracted to obtain the gravity disturbance (Torge et al. (2023))

$$\delta\mathbf{g}^n = \mathbf{g}^n - \boldsymbol{\gamma}^n = \ddot{\mathbf{r}}^n - \mathbf{C}_s^n \mathbf{f}^s + (2\boldsymbol{\Omega}_{ie}^n + \boldsymbol{\Omega}_{en}^n)\,\mathbf{v}^n - \boldsymbol{\gamma}^n \,, \tag{5}$$

where, $\boldsymbol{\gamma}^n$, in this case represent the gravity field of the Geodetic Reference System 1980 (Moritz (2000)).

## 3.2 Data Processing

The equations presented in the previous section form the basis for the data processing presented in this section. To derive gravity (disturbance) estimates, raw data from one gravimeter and one GNSS receiver is needed. This leads to six possible sets of gravity estimates, which are all made available via the data link. The overall processing flowchart is presented in Figure 5, which is constrained to a single GNSS receiver, but illustrates solutions from the two different gravimeters. The initial processing steps are different for the two gravimeters, but otherwise identical, while the GNSS data processing flow is independent of the chosen receiver.

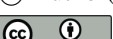

The processing strategy applied here is similar to the one described in Johann et al. (2019) and focuses on scalar gravimetry, meaning that only the vertical component of (5) will be used, i.e.

$$\delta g = \ddot{h} - f_{\mathrm{Down}} + E - \gamma = \ddot{h} - f_{\mathrm{Down}} + 2\omega_{ie} v_E \cos\phi + \frac{v_N^2}{R_N + h} + \frac{v_E^2}{R_E + h} - \gamma \,, \tag{6}$$

where $h$ is the ellipsoidal height, $f_{\mathrm{Down}}$ is the vertical component of specific force, $\omega_{ie}$ is the Earth rotation rate, $\phi$ is the geodetic latitude, $v_N$ and $v_E$ are the north and east velocities, respectively, and $R_N$ and $R_E$ are the Earth radii of curvature. The raw observations from the instruments are not directly the terms in the above expression. We must therefore first apply some pre-processing steps as indicated in Figure 5, which are done using proprietary or commercial software products.

For the raw GNSS observations, the Waypoint commercial post-processing software suite from Hexagon / NovAtel is used to derive two data products:

1. A purely GNSS based trajectory, $\mathbf{r} = (\phi, \lambda, h)$, based on a Precise Point Positioning (PPP) processing strategy. The final satellite ephemeride products made available by the International GNSS Service (IGS) are utilized. More specificically, the 5 second clock products and 30 second orbit products from the Center for Orbit Determination in Europe (CODE) are used.

2. The GNSS-based PPP solution is merged with the iMAR iNAT-RQH accelerations and angular rates to derive an integrated IMU/GNSS navigation solution consisting of attitude, velocity and position estimates. From these, the roll and pitch angles ($\alpha$ and $\beta$) along with the north and east velocities ($v_N$ and $v_E$) are needed in the processing chain.

From the GNSS-based PPP solution, the ellipsoidal height is numerically double-differentiated to obtain the kinematic acceleration, $\ddot{h}$. It is important not to use height estimates from the integrated IMU/GNSS solution to obtain these accelerations. For all other terms in (6), the integrated solution can be used.

The roll and pitch angles of the integrated solution is used to project the specific force accelerations, $(f_x, f_y, f_z)$, measured by the iMAR strapdown gravimeter onto the vertical axis as (Glennie et al. (2000)):

$$f_{\mathrm{Down}} = f_z \cos\alpha \cos\beta - f_x \sin\beta + f_y \sin\alpha \cos\beta \,. \tag{7}$$

The raw observations of the GIRAFE quantum gravimeter consists of a transition probablity from the cold-atom interferometer and a specific force obtained from a Honeywell QA accelerometer located below the vacuum chamber (Lawrence (1998)). These two observations are merged in a hybridization algorithm described in Bidel et al. (2018), to arrive at a specific force, $f_{\mathrm{Down}}$, which is directly measured along the vertical direction due to the mechanical platform. In other words, the mechanical platform assures that $\alpha = \beta = 0$, such that $f_{\mathrm{Down}} = f_z$.

Finally, before proceeding, these observables, $f_{\mathrm{Down}}$ and $\ddot{h}$, are subjected to an initial filter with a kernel width of one to two seconds. During processing this was seen to remove some error effects and produce more consistent results among the





**Figure 5.** Processing flowchart. In the top are instruments generating raw observations in the form of probability, $P_2$, vertical specific force, $f_{\text{Down}}$, sensor frame specific force, $\mathbf{f}^s$, and sensor frame rotation rates, $\boldsymbol{\omega}_{is}^s$. Then follows initial processing using proprietary software to hybdridize quantum observations for accelerations, $f_{\text{Down}}$, and commercial software to produce both a pure GNSS-derived navigation solution, $\mathbf{r}^n$ and $\mathbf{v}^n$, along with an integrated INS/GNSS solution for roll and pitch angles, $\alpha$ and $\beta$. Following these initial steps, all variables are acquired to exercute the processing described in section 3.2.

various solutions. For this, an implementation of a Resistor-capacitor Circuit (RC) low-pass filter was employed. The process is denoted as $FRC$ in Figure 5.



Following these pre-processing steps, we obtain all terms in (6), allowing us to compute the gravity disturbance. It should be noted that the specific force, $f_{\text{Down}}$, can be derived from either the GIRAFE quantum gravimeter or the iMAR iNAT-RQH gravimeter and the kinematic acceleration term, $\ddot{h}$, can be derived from any of the three GNSS receivers onboard the aircraft. The magnitude of the gravity signal is typically on the order of 10 mGal, whereas the observation noise on $f_{\text{Down}}$ and $\ddot{h}$ is typically of the order of $10^5$ mGal, meaning that a low-pass filter must be applied in order to reduce high-frequency noise on the observations. For this dataset, the final gravity estimates are supplied both non-filtered and at three different filter lengths. The applied low-pass filter is a zero-phase second order Butterworth filter with filter width specified in terms of the Full-Width-Half-Maximum (FWHM) of the kernel (Jensen (2022)).

## 3.3 External Gravity Observations

Following the processing described in the previous section, the gravity disturbance estimates derived from the iMAR strapdown sensor are corrupted by the presence of systematic errors in the gravity sensor. External gravity observations, known as tie values, are performed on the apron next to the aircraft parking position. Comparing these tie values with the gravity estimates before and after flight, while the aircraft is stationary on the apron, allows for the estimation of an offset and linear trend, which is used to correct the gravity estimates. The tie values used for this correction are listed in Table 2. Notice that the GIRAFE gravity sensor is not subject to such systematic errors and thus requires no tie value correction.

**Table 2.** External gravity observations used as tie values to estimate bias and trend of the gravity estimates derived using the iMAr strapdown gravimeter. All gravity values refer to ground level on the apron and is listed as gravity vector magnitude and gravity disturbance. Coordinates are geodetic with respect to the WGS84 ellipsoid.

| ID | Description | Latitude [°] | Longitude [°] | Height [m] | $g$ [mGal] | $\delta g$ [mGal] |
|---|---|---|---|---|---|---|
| AEY | Akureyri airport | 65.6528239422 | -18.0751787968 | 67.439 | 982337.418 | 24.240 |
| RKV | Reykjavik domestic airport | 64.1315813460 | -21.9466651338 | 76.642 | 982263.301 | 59.058 |
| UAK | Narssassuaq airport | 61.1616456551 | -45.4187238948 | 66.880 | 981923.639 | -63.743 |
| GOH | Nuuk airport | 64.1908984290 | -51.6756369418 | 109.422 | 982173.440 | -24.917 |

## 3.4 Internal Cross-Over Evaluation

As a means of internally evaluating the precision of the obtained results, gravity estimates at the same point in space resulting from two different flight segments can be compared. As illustrated in Figure 6, the airborne survey is typically designed as parallel flight tracks with perpendicular tie lines for evaluating the survey. At the points in space where the lines cross, two estimates of the gravity value are available and can be compared. If a considerable number of such cross-over points are available, it is possible to derive some statistics from the differences as illustrated in Figure 6. The most commonly quoted value is the Root-Mean-Square-Error (RMSE), defined as



$$\text{RMSE} = \frac{\text{RMS}}{\sqrt{2}} = \sqrt{\frac{1}{2N}\sum_{n=1}^{N}\epsilon_i^2}\,, \qquad (8)$$

where $\epsilon_i$ denotes the cross-over difference and $N$ is the total number of crossing points. The RMSE value is typically associated with the survey accuracy.

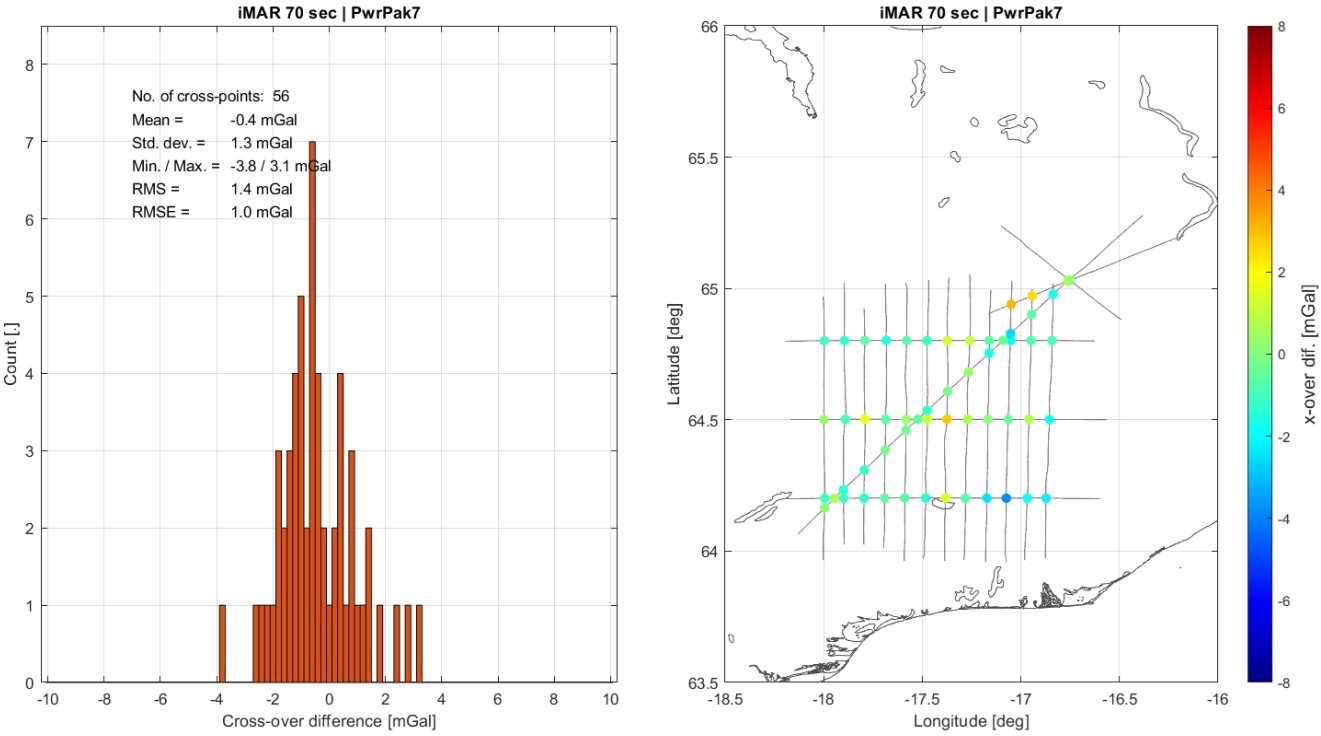

**Figure 6.** Illustration of cross-over error evaluation. The figure title indicates gravimeter (iMAR), GNSS receiver (PwrPak7) and filter width (70 seconds). (Right:) Cross-over differences in estimated gravity values from two crossing flight lines; (Left:) Histrogram derived from the cross-over differences. Also shown is the statistics derived from the differences.

It should be noted that crossing flight lines may not be carried out at the same flight altitude. Since the normal gravity field is removed from the full gravity magnitude, a vertical variation corresponding to that of the GRS80 normal gravity field is accounted for. In practice, the vertical variation may differ from that of the model and care much be taken when comparing estimates obtained at significantly different heights.

### 3.5 Comparison with other Gravity Observations

Another independent evaluation is to compare new gravity observations with existing gravity observations in the area. Generally, these observations are collected both at sea, on land, on ice and in the air, meaning they must be interpolated and





extrapolated both horizontally and vertically in order to evaluate the new observations. Both Iceland and Greenland have nu-
merous sources of gravity anomaly data in the joint Nordic gravity data base, both onshore and offshore. Although many land
data sources date back to the 1950's, all data have been rectified and quality checked to modern gravity standards, and quality
checked extensively in connection with recent Greenland and Iceland Geoid projects. Since the existing data are available in
the form of gravity anomalies, $\triangle g$, rather than gravity disturbances, $\delta g$, used in this paper, the upward continued ground data
are converted to gravity disturbances.

The upward continuation process has been done by Fast Fourier transformation in a remove-restore process, taking into
account both a Global Gravity Model (GGM) and terrain effects. The GRAVSOFT suite of programs (Forsberg and Tscherning,
2014) have been used in this process, in the following steps:

1. Reduction of the gravity anomalies, $\triangle g$, by computing a reference field, $\triangle g_{\text{GGM}}$, using the EIGEN-6C4 model to degree
   and order 180 (Förste et al. (2014)) and forming the residuals (GRAVOSFT `geocol17`):

$$\triangle g_{\text{GGM}} = \triangle g - \triangle g_{\text{GGM}} \,.$$

2. Removing the short-wavelength topographic effects using Residual Terrain Modelling (Forsberg (1984)) by computing
   the gravitational attraction, $\triangle g_{\text{DEM}}$, from topography, bathymetry and ice mass using prism integration (GRAVOSFT
   `TC`):

   $$\triangle g_{\text{GGM+DEM}} = \triangle g_{\text{GGM}} - \triangle g_{\text{DEM}} \,.$$

3. Gridding the reduced gravity data in a regular grid using least squares collocation (GRAVOSFT `GEOGRID`) and then
   upward continuing gravity data to flight altitudes (GRAVOSFT `GEOFOUR`), taking into account the differing flight ele-
   vations using 3D "sandwich" interpolation between several flight levels:

   $$\triangle g_{\text{GGM+DEM}}^{H} = \text{FFT}\left[\triangle g_{\text{GGM+DEM}}\right]$$

4. Restoring the GGM and terrain effects at the flight altitudes (GRAVSOFT `TC` and `GEOCOL17`), and converting the
upward continued gravity anomalies to gravity disturbances:

   $$\delta g^{H} = \triangle g_{\text{GGM+DEM}}^{H} + \triangle g_{\text{DEM}}^{H} + \triangle g_{\text{GGM}}^{H} + 0.3086 \cdot N \,,$$

   where $0.3086\,\text{mGal/m}$ is the vertical gradient of gravity and $N$ is the Geoidal undulation, i.e. the height of the Geoid
   above the ellipsoid.

Following these steps, the computed gravity disturbances are mildly filtered to match approximately the inherent along-track
filtering in the airborne data.

### 3.5.1 Iceland

The gravity data of Iceland covers both the ice free land areas, the major glaciers, as well as numerous sources offshore, illustrated in Figure 7. The DEM and gravity data in the data base of the Nordic Geodetic Commission (NKG) have been provided by Landmælingar Islands (Iceland Geodetic Survey), with data on the ice, including radar echo sounding of ice thickness, provided by the University of Iceland. The terrain effect computations take into account the density of ice (0.92 g/cm$^3$) is different from the conventional rock density of 2.67 g/cm$^3$. The land DEM of Iceland is based on 2" resolution data, while no bathymetry data has been used. The Geoid model used for anomaly to disturbance correction are based on the latest Iceland geoid model (Forsberg and Valsson, unpublished). Due to the relatively dense gravity data coverage around the flight tracks, the error of the upward continued gravity disturbances are estimated at 2-3 mGal.



**Figure 7.** Illustration of existing gravity data in Iceland along with the flight lines of the current campaign.





### 3.5.2 Nuuk

In the Nuuk fjord system the large fjords, many of which unsurveyed have depth well in excess of 1000 m, meaning the fjord itself generates narrow gravity anomalies up to 60-80 mGal, not captured well by the 1960's numerous land gravity points along the fjords. With the lack of fjord depths, the sparse bathymetry available was not used, but instead existing high-altitude airborne gravity data US Naval Research Lab 1991-92 (Brozena (1992)), from Operation IceBridge (2009-12), as well as low-level airborne data from DTU Space (2003-7) at low level along the offshore region was used. To utilize all data together, advantage was taken from the 2016 Greenland Geoid model (GGEOID16, see Forsberg and Jensen (2016), and the International Service for the Geoid, www.isgeoid.polimi.it). Here all gravity data available in Greenland and the surrounding oceans were combined in a rigorous downward continuation to the surface, using blocked least squares collocation (GRAVSOFT `gpcol`), taking into account the different accuracies and elevations of the various data sources. The GGM was EIGEN-6C4 as in Iceland, with DEM surface data taken from Byrd Polar Research Center (BPRC 2013) from optical imagery, downsampled to 500 m resolution, with ice thickness from Bamber et al. (2001), regridded to the DEM spacing. Due to the lack of airborne radar data over the margins of the Greenland ice sheet, the ice thickness model have large errors here (of less importance for the Nuuk flight region). For the intercomparison with the airborne gravity data presented in this manuscript, the downward continued gravity anomaly grid at 2 km resolution from the GGEOID16 computation was upward continued to the 2023 flight elevations, and converted to gravity disturbance. The data coverage underlying the GGEOID16 is shown in Figure 8.

### 3.6 Combining Estimates from Two Sensors

In the results section there will be indications of increased spatial coverage and resolution from the iMAR strapdown sensor, while the GIRAFE system shows superior results in terms of stability. As these properties are complementary, this motivates a combination of the two sensors. An in-depth study of the hybridization of these sensors is outside the scope of this manuscript, but a simple combination approach will be used to illustrate the potential and some characteristics of the data.

The combination is performed on a line-by-line basis by forming the Fast Fourier Transform (FFT) of the estimates coming from each instrument. This leads to two sets of coefficients

$$
\begin{aligned}
X_k^{\text{iMAR}} &= \textbf{FFT}\left[dg^{\text{iMAR}}(x)\right] \\
X_m^{\text{GIRAFE}} &= \textbf{FFT}\left[dg^{\text{GIRAFE}}(x)\right]
\end{aligned}
\tag{9}
$$

where $k = 0, 1, ..., K$ and $m = 0, 1, ..., M$ are the number of coefficients for each gravity profile and $x$ refers to the along-track distance. In order to retain the high-frequency information (spatial variability) from the iMAR instrument and the low-frequency information (stability) from the GIRAFE instrument, we can use the low-degree coefficients from GIRAFE and higher-degree coefficients from iMAR. We may therefore form a new set of coefficients

$$
X_n^{\text{comb}} = \left\{ X_1^{\text{GIRAFE}}, X_2^{\text{GIRAFE}}, ..., X_N^{\text{GIRAFE}}, X_{N+1}^{\text{iMAR}}, X_{N+2}^{\text{iMAR}}, ..., X_K^{\text{iMAR}} \right\} ,
\tag{10}
$$





**Figure 8.** Illustration of existing gravity data in the Nuuk area along with the flight lines of the current campaign.

from which we can restore the gravity profile using an inverse FFT transform to arrive at a combined estimate:


$$dg^{\mathrm{comb}}(x) = \mathbf{iFFT}\left[X_n^{\mathrm{comb}}\right] . \tag{11}$$

The amount of information retrieved from the GIRAFE and iMAR estimates can be varied by changing the degree, $N$, indicating the number of coefficients used from the GIRAFE estimates.

## 4 Results and Discussion

This section aims to give a brief overview of the results and point out some key points. The results will be divided into four

sections, defined by geographical location.



### 4.1 Vatnajökull Ice Cap and Askja Volcano

Gravity estimates over the Vatnajökull Ice Cap are illustrated in Figure 9. To the North-East of the ice cap is the Askja volcano, which has been associated with significant vertical displacement (Pagli et al. (2006)). Two sets of gravity estimates are illustrated; one derived using the GIRAFE quantum instrument and one derived using the iMAR strapdown gravimeter.

From the figure it is evident that the iMAR dataset has a better spatial coverage than the GIRAFE dataset. This is the result of a subjective process where some data is discarded by the person processing the data, mostly related to aircraft turns, but also as a result of turbulence or other artefacts. The improved spatial coverage has been observed previously and is most likely related to the mechanical platform rather than the sensor technology (Jensen et al. (2019)). On the other hand, the platform has also been seen to provide better long-term stability of measurements.

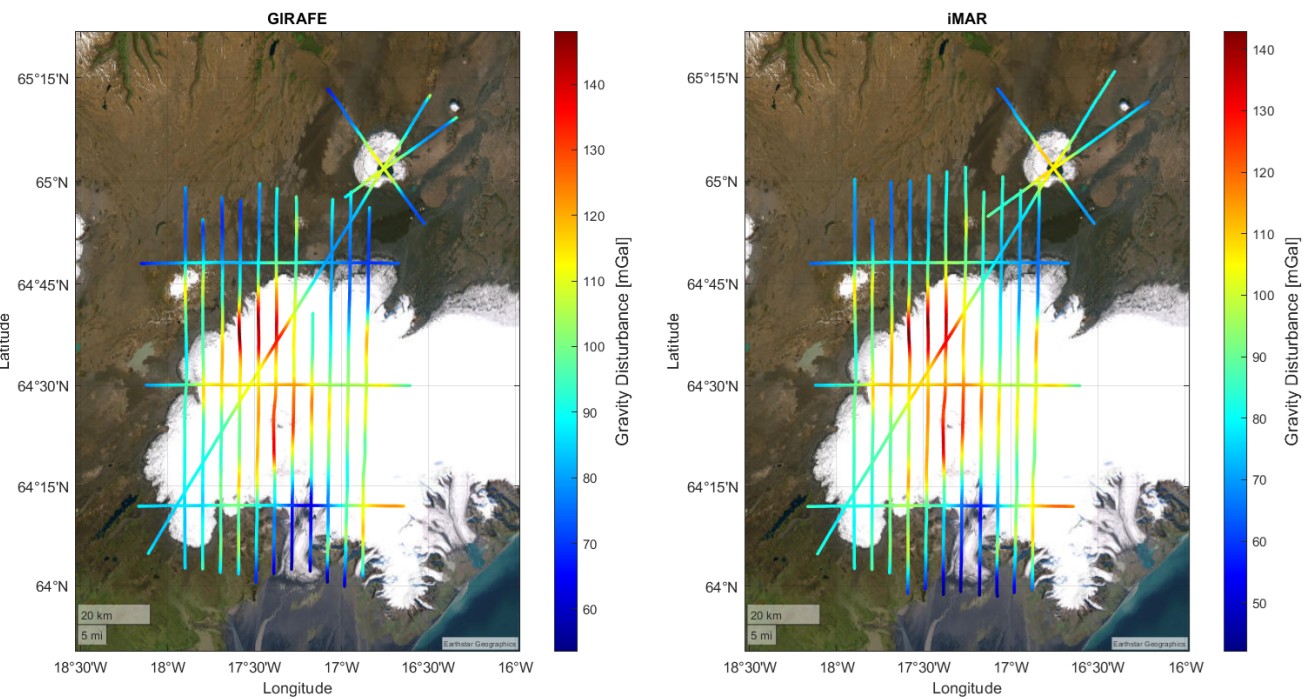

**Figure 9.** Gravity disturbance estimates from Vatnajökull Ice Cap and Askja Volcano. Kinematic accelerations are derived from the NovAtel PwrPak7 GNSS receiver and disturbance estimates are low-passed filtered using a FWHM filter width of 70 seconds. (Left:) Estimates derived from GIRAFE; (Right:) Estimates derived from iMAR.

The two datasets in Figure 9 are not unique. They are derived using one of the three available GNSS receivers. Furthermore, the results depend on the final low-pass filter applied to the gravity disturbance estimates. As a way of objectively investigating the optimal amount of filtering, some statistical measures can be derived from the cross-over differences using several values of the filter width (Full-Width-Half-Maximum). Such an analysis has been carried out in Figure 10, indicating that a value of



around 70 seconds for the FWHM provides optimal results with an accuracy of around 1 mGal for both systems in terms of the
RMSE value.

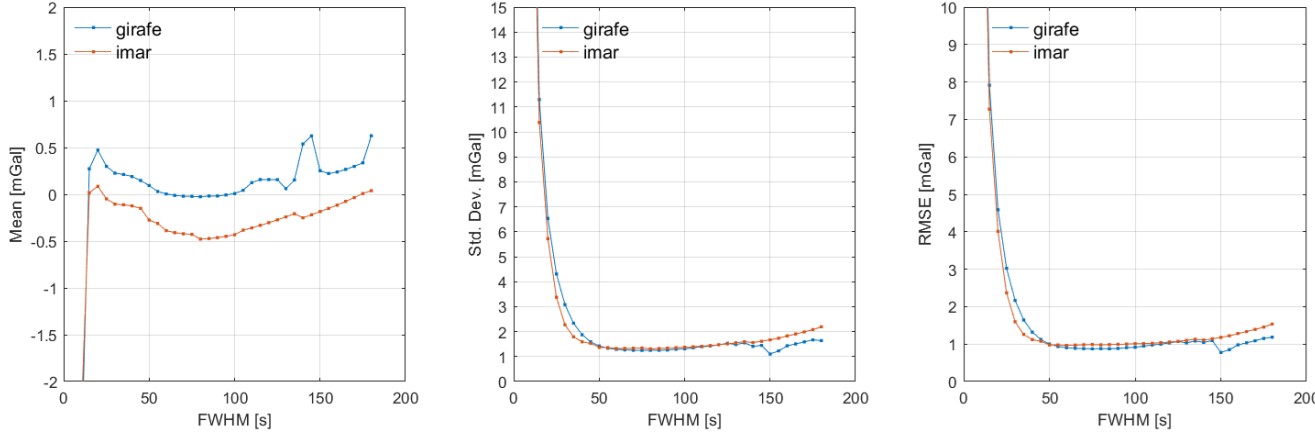

**Figure 10.** Mean, standard deviation and RMSE as a function of filter width derived from cross-over differences at Vatnajökull Ice Cap and
Askja Volcano.

Interpretation of the mean value to the left in Figure 10 is not straightforward. However, if systematic errors, other than a
constant offset, is present in the data, such errors would lead to non-zero mean values. The figure may therefore be an indication
of the improved stability of the GIRAFE quantum instrument, while significant systematic errors may be present for the iMAR
strapdown instrument.

**4.2    Fagradallsfjall Volcano**

The Fagradallsfjall Volcano, situated only 35 km from Keflavik International Airport, has seen a number of recent eruptions,
with one occuring just a few days after our survey (Halldórsson et al. (2022); Sigmundsson et al. (2022)). Such volcanic
eruptions indicate a direct re-distribution of mass that will affect the associated gravity field. Figure 11 illustrates both the
measured and predicted gravity values along the flight lines of the current campaign.
Although the order of magnitude are the same and some spatial patterns correlate, it is difficult to directly compare the
two signals. Inspecting Figure 7 it is evident that the spatial coverage of previous observations is relatively poor in the area.
As a result, the detailed signal is mostly dictated by the DEM used for the area, which again may not be accurate since the
topography has actively changed due to deposition of lava on the surface. This newly acquired data may thus provide useful
information in future studies of mass transportation related to volcanic activity in the area.



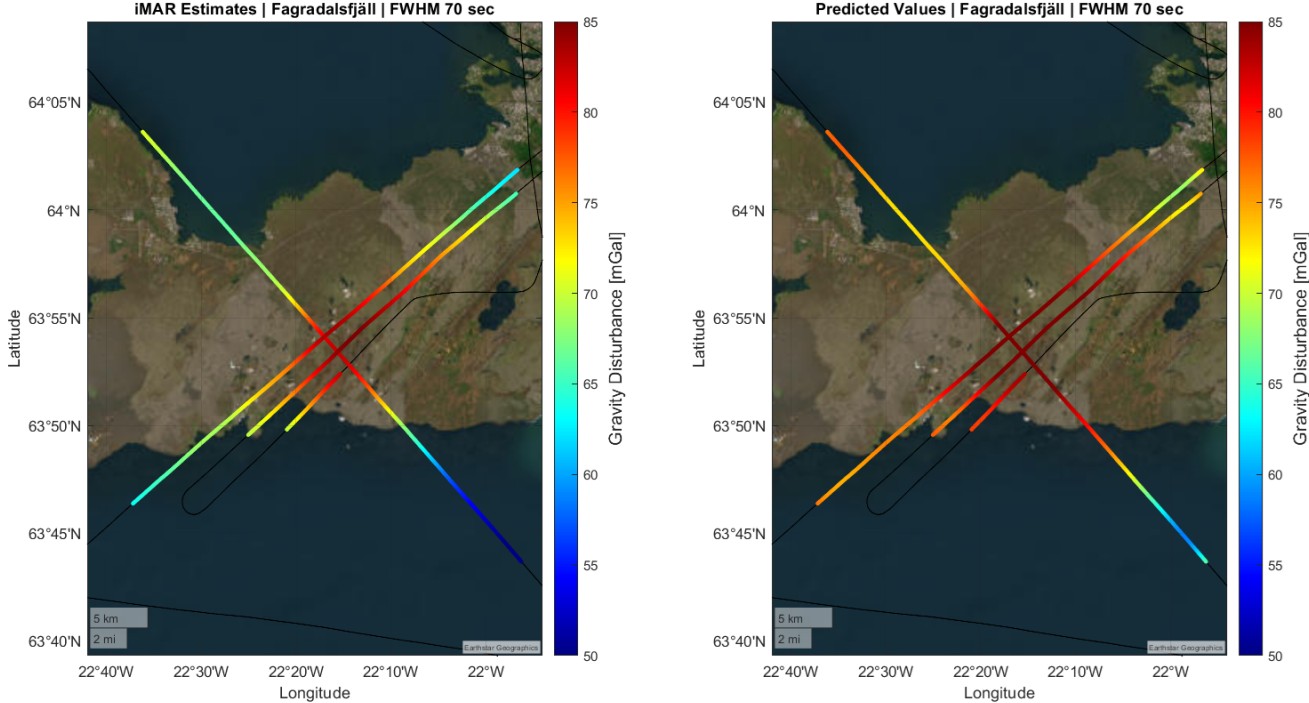

**Figure 11.** Gravity disturbance estimates from: (Left:) iMAR strapdown sensor from new airborne survey; (Right:) Predicted gravity disturbance from previous gravity observations.

## 4.3 Flight Line from Snæfellsjökull to Akureyri

During the previous flight campaign in 2017, a flight line between Akureyri and Snæfellsjökull was surveyed from westward and eastward using the GIRAFE instrument, see Bidel et al. (2020). During the 2023 campaign this survey line was flown again during transit from Greenland back to Akureyri. The resulting gravity estimates are shown in Figure 12 along with the estimates from the 2017 campaign and those predicted from the existing gravity observations.

The figure shows that the new gravity estimates correlate quite well with the predicted estimates, although some significant differences are apparent during the first 250 km of the line. Again, referring to Figure 7 it is evident that this correlates with poor data coverage, only north of the parallel mountain ridge, and the crossing of a fjord with no associated data.

The bottom figure shows the difference with the 2017 campaign, which was the first ever flight test of the quantum gravimeter. This figure is an indication of the improvement that the instrument has undergone during those six years of development.

## 4.4 Nuuk Fjord System

Gravity estimates from the Nuuk fjord system is shown in Figure 13. By comparing the GIRAFE and iMAR datasets it is again evident that the strapdown system results in a superior spatial coverage. The mean, standard deviation and RMSE values



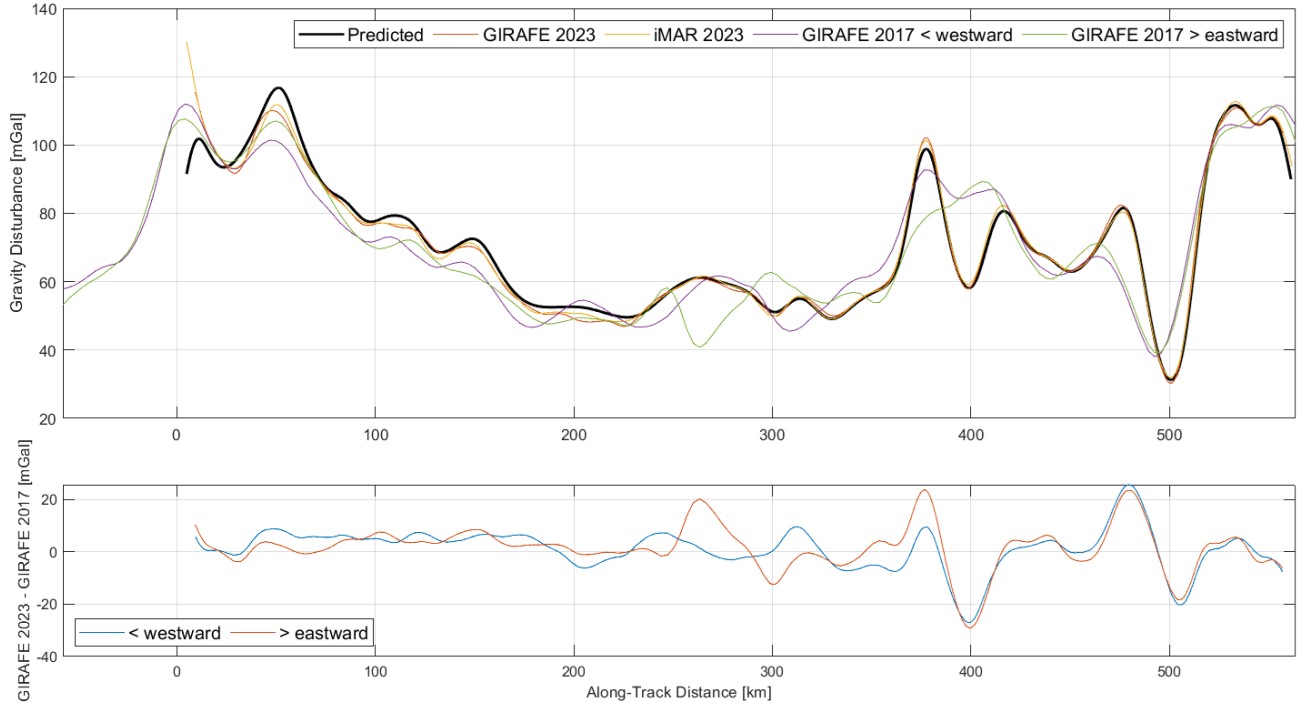

**Figure 12.** Gravity disturbance estimates along flight line from Snæfellsjökull to Akureyri, notice that this line was flown once from west to east during the campaign in 2023, while it was flown twice during the campaign in 2017. (Top:) Predicted from existing observations (black), from GIRAFE 2023 (red), iMAR 2023 (yellow), GIRAFE 2017 (purple and green); (Bottom:) Difference between GIRAFE estimates in 2023 and 2017.

derived from cross-over differences are shown in Figure 14, indicating an accuracy of around 1.5 mGal - 2.0 mGal in terms of RMSE and an optimal value of 40 seconds for the filter FWHM.

It is generally expected that noise attenuates with filtering, such that more filtering leads to better results. However, the filtering not only removes noise, but also smooths out the actual gravity signal as the aircraft moves along its path. This means that the derived gravity estimate represents an average gravity value along the aircraft trajectory, depending on the shape of the filter kernel and the filter width. In a cross-over analysis, the flight lines are not parallel, meaning that gravity estimates represent averages over different trajectories in space. For this reason, the derived RMSE value is expected to increase when

the error induced by smoothing the signal along different directions becomes larger than the noise reduction due to filtering. An alternative to cross-over validation is to repeat the same line segment twice. In Nuuk, this was done with the East-West tie line, crossing the sixteen north-south oriented parallel lines, which was flown twice. In this case we can directly form the difference in gravity estimates along the line and derive statistics similar as with the cross-over differences. This is shown in Figure 15.

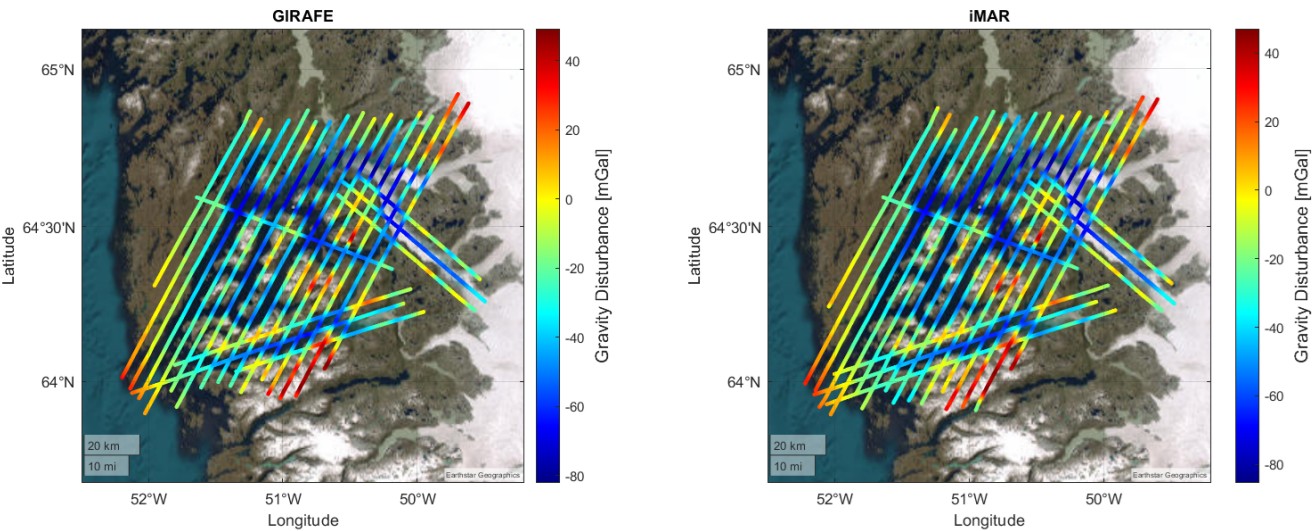

**Figure 13.** Gravity disturbance estimates from Nuuk fjord system. Kinematic accelerations are derived from the NovAtel PwrPak7 GNSS receiver and disturbance estimates are low-passed filtered using a FWHM filter width of 40 seconds. (Left:) Estimates derived from GIRAFE; (Right:) Estimates derived from iMAR.

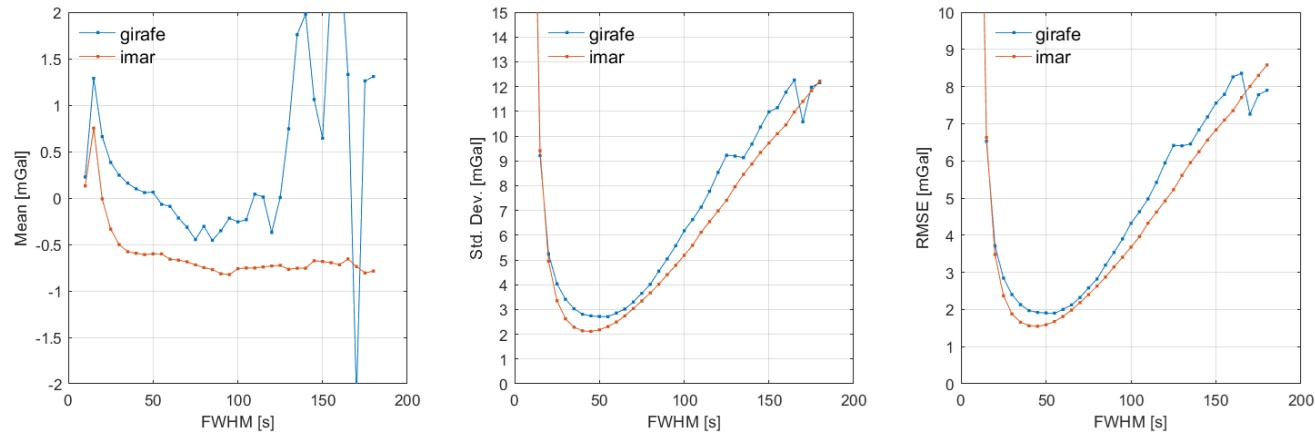

**Figure 14.** Mean, standard deviation and RMSE as a function of filter width derived from cross-over differences at Nuuk fjord system. Notice that as the filtering increases fewer cross-over points become available, leading to unstable statistics.

Inspection of the mean value in Figure 15 indicates a small offset between the GIRAFE estimates, while a significant offset is present for the iMAR estimates. The standard deviation and RMSE values are seen to generally decrease as a function of more filtering. As argumented above, this is as expected. One twist is that erroneous estimates from aircraft turns will increasingly propagate into the straight line segments with increased filtering. This is handled by removing more and more from the beginning and end of line. As a result, the length of the line that can be compared decreases with increased filtering.

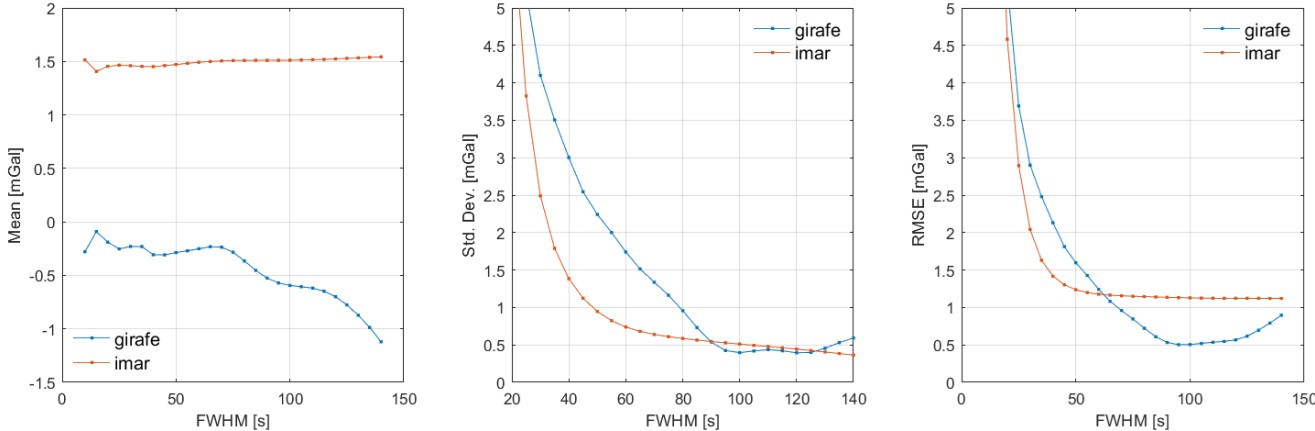

**Figure 15.** Mean, standard deviation and RMSE as a function of filter width derived from differences between two repeats of the east-west tie line crossing all the north-south parallel lines.

The filtering will also not remove systematic errors such as bias. This is evident by comparing the standard deviation and RMSE values for the iMAR estimates. Since the standard deviation accounts for a mean value, this value drops to around 0.5 mGal, while the RMSE value flattens out at around $1.5/\sqrt{(2)} \approx 1.1$ mGal, corresponding to the mean difference of 1.5 mGal.

Comparing Figure 14 with Figure 10 indicates that less filtering is required in Nuuk before the errors induced by spatial filtering surpasses the noise reduction from filtering. Theoretically, this is in line with expectations since the gravity field in Nuuk varies much more in terms of magnitude and distance compared with the Vatnajökull Ice Cap. This is a direct consequence of the topography resulting from deep fjords and steeps mountains slopes.

### 4.5 Combining Estimates from Two Sensors

To illustrate the potential of hybridizing the two sensor technologies, the simple combination strategy presented in section 3.6 is used to arrive at combined estimates. Such hybridization is relevant also for the definition of future satellite missions. Figures 16 and 17 shows the mean and RMSE values derived from a combination of the two instruments. The blue and red curves are the same as those presented in Figures 10 and 14.

The two figures illustrate that the combination works according to the intention. First of all, the mean value of the GIRAFE estimates are adopted, whereas the lower standard deviation of the iMAR estimates are adopted. As a result, the combined accuracy in terms of RMSE decreases to around 0.5 mGal for Vatnajökull and 1.2 mGal for Nuuk. Additionally, we may notice that no improvement follows from using more than the zero degree term from the GIRAFE estimates. This may indicate that the iMAR estimates are subject to a line-by-line bias error.



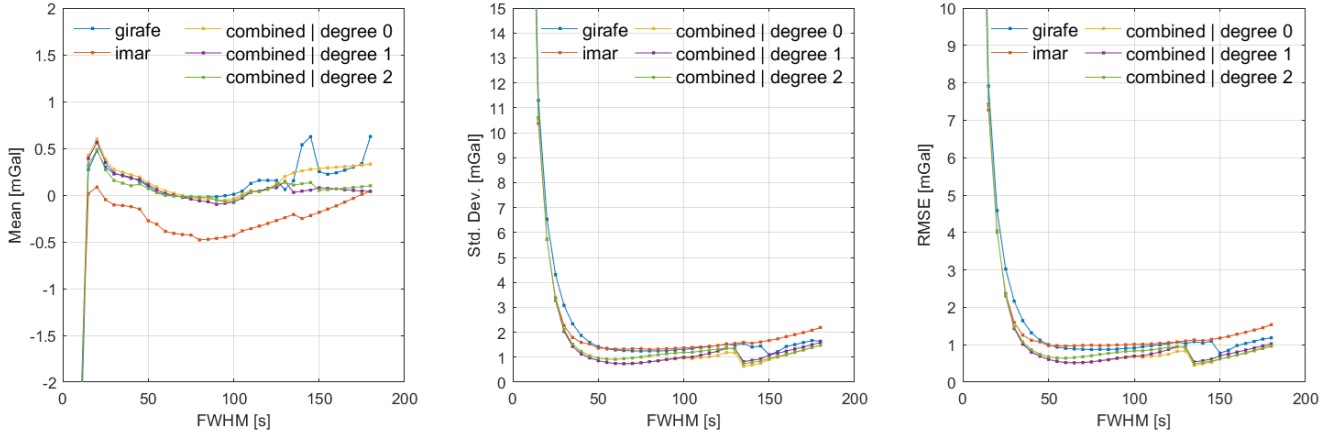

**Figure 16.** Mean, standard deviation and RMSE values derived from cross-over differences at Vatnajökull Ice Cap. These statistics are derived as a function of filtering for both GIRAFE estimates (blue), iMAR estimates (red) and a combination of the two instruments. The three different variants represents results using a: only the zero degree term from GIRAFE (yellow), the zero and first degree term from GIRAFE (purple) and the zero, first and second degree term from GIRAFE (green).

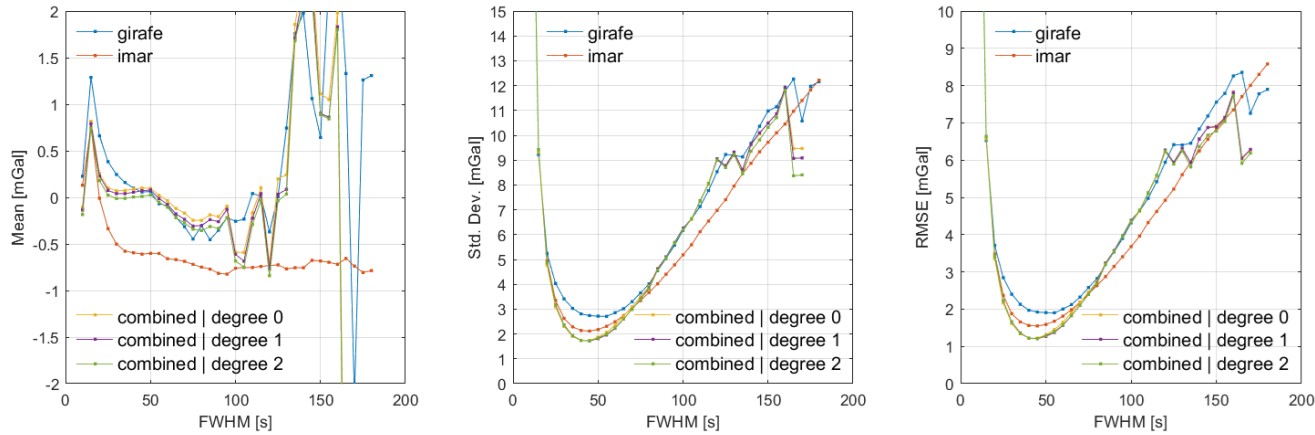

**Figure 17.** Mean, standard deviation and RMSE values derived from cross-over differences at Nuuk Fjord System. These statistics are derived as a function of filtering for both GIRAFE estimates (blue), iMAR estimates (red) and a combination of the two instruments. The three different variants represents results using a: only the zero degree term from GIRAFE (yellow), the zero and first degree term from GIRAFE (purple) and the zero, first and second degree term from GIRAFE (green).

## 5 Conclusions

This manuscript presents a dataset obtained from an airborne gravity campaign covering the Vatnajökull Ice Cap along with two volcanic targets in Iceland and the Nuuk Fjord System in Greenland. The instrumentation includes the GIRAFE cold-atom





gravity sensor and the iMAR iNAT-RQH strapdown gravity sensor along with three different GNSS receivers. The data from
the campaign is made available in three categories, depending on the amount of post-processing applied. See the section on
data availability. This should enable anyone to use the data, whether interested in investigating processing techniques or using
the data for geophysical studies.

Following an overview of the survey and instrumentation, the processing strategy applied to derive level 1 and level 2 data
products is presented in the manuscript. Finally, some results are presented in order to describe the characteristics of the data
and show the potential for using the data in geophysical studies.

## 6   Data availability

The data from the two campaigns are made available via the ESA Earth Online webpage (Jensen et al. (2024), https://doi.
org/10.57780/esa-58c58c5). The data is divided into raw, intermediate and final data products, depending on the level of post-
processing applied. An overview is given in Table 3. A more elaborate description of the data products and format can be found
in the associated ESA report and the README files distributed along with the data.





**Table 3.** Description of the data products made available from the campaigns

| **Level 0** |
| --- |
| • Raw GNSS data from the GNSS receivers (RINEX) |
| • Time-tagged accelerations and angular rates from the GIRAFE stabilized platform (ASCII) |
| • Time-tagged accelerations from the GIRAFE cold-atom accelerometer (ASCII) |
| • Time-tagged accelerations and angular rates from the iMAR strapdown IMU (ASCII) |
| **Level 1** |
| • Processed GNSS data with estimates of position and velocity derived from PPP processing. These estimates refer to the position of the GNSS antenna (ASCII). |
| • Kinematic accelerations derived from GNSS position estimates at the three positions (ASCII):<br>1. GNSS antenna<br>2. iMAR strapdown sensor<br>3. GIRAFE accelerometer |
| • Time-tagged navigation profile (position, velocity and atitude) and specific force accelerations along the plumb line direction from the two gravity sensors (ASCII) |
| **Level 2** |
| • Time-tagged and geo-located gravity estimates from the GIRAFE instrument (ASCII) |
| • Time-tagged and geo-located gravity estimates from the iMAR instrument (ASCII) |
| • Upward continued terrestrial observations at the location of the gravity sensors (ASCII) |

*Author contributions.*

TEJ and RF designed the survey lines. TEJ coordinated the project and YB coordinated the ONERA contribution. All authors participated in the field campaign. TEJ carried out the data processing with support from YB. TEJ performed the data analysis

and wrote the manuscript. RF performed the upward continuation of external data.

*Competing interests.*

The contact author has declared that none of the authors has any competing interests.



*Acknowledgements.* We thank Danish Defence Acquisition and Logistics Organisation (DALO) and the European Space Agency (ESA) for
support of the airborne campaign. The project was also supported by Landmälingar Islands and University of Iceland by providing data to
the NKG Nordic gravity data base and ice thickness data. The survey flights was carried out and made possible by the support of Norlandair,
Akureyri. The authors would also like to acknowledge the internal financial support of DTU and ONERA that made this study possible.



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
