# Peer review of "Airborne Gravimetry with Quantum Technology: Observations from Iceland and Greenland"

_Earth System Science Data, 2024_

## Author Response (AR1)

**List of Corrections**

**Detailed comments from reviewer 1:**

Line 11. "..from \*an\* airborne gravity campaign..." Corrected

L14. Suggest adding "...only measure the \*relative\* variation of gravity...". And technically, by "gravity" you should point out that you at least mean "acceleration of gravity" or, preferably, "specific force". This comes up later, but it's probably best to be precise from the beginning. Corrected

L62. Without knowing which way north is, it is not so easy to identify the "east-west flight line". Recommending adding a "North symbol" to the figure. North arrow added to figures 1 and 2

L70. In addition, the GIRAFE should have other advantages, true?: No need for airport ties, each measurement is independent of previous measurements (neglecting any co-sensors), etc. The following text was added: "traditionally, an airborne gravity survey requires calibrations before and after each flight, thus depending on external gravity values from either an existing gravity network or additional absolute gravity measurements performed at the airport. The independence of such an infrastructure is a major operational advantage"

L90. For completeness, is it possible to report on the distance from the antenna (and thus gravimeters) to the aircraft's center of mass? Unfortunately we do not have this information.

L133. Perhaps explicitly state that delta\_g is the gravity disturbance here (rather than "introduce" it in the next section..). Expression added in text

L158. Now I am intrigued: can you say more about not using the IMU/GNSS solution to determine h\_doubledot (but it \*is\* okay to use it for h)?

The INS navigation solution is based on acceleration observations, meaning that any variation of the gravity field not corrected for will influence the navigation solution. Since you want to separate the kinematic acceleration from gravity acceleration, you don't want to "corrupt" your kinematic accelerations. To summarize: GNSS measures accelerations only from movement, IMU measures accelerations from movement and gravity. I hope this makes sense.

L162. This is the first we have seen (x,y,z) components. At this point (probably earlier where different inertial frames are first mentioned...) we need a figure at least linking xyz to alpha and beta.

I think I have to disagree with this point. Maybe this is due to background and working environment. A figure of reference frames just seems a bit too elementary. It can be found in many elementary book on the subject and I don't think such a figure is needed in a scientific publication.

L176. This sentence is a bit confusing. Suggest something like "magnitude of the expected gravity signal variation at altitude..." along with "... a heavy low pass filter..." emphasizing that

the SNR is tiny. done

Table 2. I suggest pointing out in the caption that delta\_g is the disturbance. It could be confused for the uncertainty on g. Note that none of these values has an associated uncertainty (and at least the gravity and disturbance really should). Also, how were the apron values determined? Using GIRAFE or a "classic" absolute meter? If the latter, was the GIRAFE in agreement at the apron site?

I wrote this out as text to clarify. The tie values were derived from previous observations at the airport and then "transferred" to the aircraft location by assuming the gravity disturbance is constant. The values was in agreement with the GIRAFE observations, but this has not been thoroughly investigated.

Figure 6. Recommend swapping the two images left for right: It is more customary to describe the left plot before the right.

This figure has been updated.

L215. The (unnumbered...) equation appears to be nonsensical: a = b - a. The residuals have the same symbol as the reference field.

The notation has been changed. The resulted in a change the last equation of this section as well.

L277. As above, recommend putting a north indicator on the map. North indicator has been added to figure 7 and 8.

Figure 10. Suggest to rewrite the caption as "Mean, standard deviation, and RMSE of the crossover differences as a function of filter width...". It's not clear at all from the figure nor until very late into the caption sentence what's actually being plotted! This has been done.

L.280. Note that the "superior coverage" of the iMar is technically true, but actually quite subtle and not obvious at first glance. Please quantify the difference. The same is true later for the Nuuk Fjords on L317.

A parenthesis had been added quantifying this difference.

L291. Re: the mean plot in Figure 10... The fact that iMar mean crossover values are almost all below 0, with the worst mean crossover actually happening at the best/chosen FWHM value (which makes sense given the minima in both stdev and RMSE...) needs more discussion. I assume it just means that one (or more) of the cross lines has a bias. Can the iMar suffer a "tare" relative the tie value at the airport? What about attempting to match a long wavelength, mean value to a global model to remove the bias? Anyway, more discussion is suggested. (Especially as this seems to be the final conclusion. See L351)

To be honest, I am not sure how to interpret this. As I write in the text, interpreting the mean of the crossovers is a not straightforward.

My impression is that a very large area is needed for the introduction of a global model. Theoretically, an area of around 200 x 200 should be adequate considered the spatial resolution of the latest satellite-only models. However, still one would have to be very careful about understanding what is going on, especially in areas with sparse data.

Section 4.2 This little section does not seem to add too much to the discussion. I understand that the ground signal is expected to have changed, but is it possible to at least quantify the

differences between measurement and prediction? One is left to "guess" at value by looking back and forth at 11a and 11b. The size of the change would be interesting to compare later with the accuracy/resolution of the instruments (for example).

I think digging more into this is a topic for other studies / investigations. The purpose of the current manuscript is merely to present the data.

L327. Again, a north indicator on the maps would help explain the discussion. Apparently the east-west repeat line is the "lonely" line in the middle (?). North indicators have been added to figure 13.

L332. There are a few small English mistakes here and there, but "argumented" should definitely be "argued". Corrected

L33. A choice of 40s seems reasonable when looking at the crossover results, but the east-west repeats would indicate something more like 90 or 100 (especially for the GIRAFE). This probably needs more discussion. Is the conclusion that east-west repeats are not a good method for estimating optimal filter length? This would be important, because in rugged terrain, one might expect that different filter lengths could "smear" a feature in one direction, and minimizing that would help in filter length choice.

Again, this is maybe digging a bit too deep for presenting the dataset. But yes, repeat lines are not good for determining the "best" filter length since you are filtering along the same direction is space. Theoretically you end up filtering out any variation and just having a bias between lines. The overall idea is to find the best compromise between filtering out actual gravity signal and noise in the observations. When the spatial direction is different you are not "averaging the same profile" whereas this is the case for a repeat line.

However, note that the intention here is to do exactly that, i.e. "finding the best compromise". You might have other purposes or desires when choosing your filter length depending on the application.

**Detailed comments from reviewer 2:**

L19, L21, etc.: According to the ESSD submission guidelines (https://www.earth-system-science-data.net/submission.html), the format for in-text citations should follow these rules:

- For example, "(Jensen(2024))" should be formatted as "(Jensen, 2024)."

- Similarly, "(Bidel et al. (2020), Bidel et al. (2023))" should be revised to "(Bidel et al., 2020; Bidel et al., 2023)."

Not sure how to correct this. I am just using the Latex template

L80: It would be beneficial to include information about the model of the GNSS antenna as well. I am unaware of the GNSS antenna model. It was preinstalled on the aircraft.

L89-90: How was the lever-arm estimated? It would be helpful for readers to include a brief explanation of the estimation method, along with a reference if available. Additionally, since the accuracy of the lever-arm affects the absolute accuracy of the observed gravity values, it would be helpful to include their estimation errors, if available.

The lever arm was initially estimated crudely with a ruler and afterwards estimated using a Kalman filter by including them in the state vector as constants. This gives an estimate for each

flight. The estimates of the two horizontal components vary by 1-2 cm, while the vertical component varies by more than one meter. This has something to do with the estimability of the states. To my knowledge nobody has really investigated this and also not the impact of errors in the estimated lever arm. This would be an interesting study. I have added a comment in the text.

L150: It would be helpful to include the version of the ITRF as well. I am afraid I am not aware of the ITRF version. The chosen reference frame in the processing software was WGS84.

Table 2 caption: "iMAr" should be corrected to "iMAR." Done

Figure 6: The title of the scale bar in the right panel should be changed from "x-over" to "crossover" for consistency. Done

Figure 6 caption: It would be more intuitive to describe the figure from left to right. This has been changed

L215, 219, 223, 226: Equation numbers should be assigned to these formulas as well. Additionally, the same symbol is used for "GGM-reduced gravity anomaly" and "GGM-derived gravity anomaly". This should be corrected. This has been corrected

L220, 248: It would be helpful to specify which covariance function model was used in these LSC calculations.

This has been added to the text

L235-236: The sentence "The terrain effect computations take into account the density of ice (0.92 g/cm3) is different from the conventional rock density of 2.67 g/cm3" is somewhat difficult to read and could be revised for better readability. This has been re-phrased.

L239: It would be helpful to include an explanation of how the error in upward continued gravity disturbances (2-3 mGal) was estimated.

A comment on this has been added to the text.

L280, 317: While it is true that the iMAR measurements cover longer survey lines, describing this as "evident" might be slightly overstated. I suggest using a more moderate expression here. This has been quantified as suggested by another reviewer

Figure 10 & 14: The cross-over difference statistics for iMAR appear to change smoothly with filter length, whereas those for GIRAFE seem to lack this smoothness. What specific characteristics of GIRAFE might contribute to this behavior? If there are any potential causes, it would be beneficial to include them in the text.

This is commented on in the caption of Figure 14, where this is worst. Essentially, it seems that the iMAR estimates are more "stable" towards the beginning and end of each flight line. The aircraft turn seems to influence / propagate much further into the gravity estimates on the straight line for the GIRAFE than for the iMAR. As a result more data is discarded for the GIRAFE than the iMAR. The statistics for the GIRAFE is therefore calculated using less cross-over points

than for the iMAR. At some point this becomes instable / meaningless. For example, the mean at FWHM = 180 sec is computed using only 6 values for the GIRAFE, but 54 points for the iMAR

L342: It would be helpful to provide an explanation of the method used to hybridize the GIRAFE and iMAR data.

I am not really sure what is meant by this? Section 3.6 is dedicated to describing this.

Figure 16 & 17: I recommend creating a table to summarize the cross-over difference statistics for iMAR, GIRAFE, and the combined solution at the optimal filter lengths. Such a table would provide a clear and concise comparison for readers.

In my opinion this contradicts the presented dataset. The data is made available both without and with several filters applied. I would prefer to keep a more open-ended decision on choosing the filter length, since this may depend on the application that someone wants to use the data for. If chosen as the minimum, the objective is to reach the best possible RMSE. But this might not represent what you want to "see" in the data.

However, I can add such a table if this is a requirement.

**Detailed comments from reviewer 3:**

A comment and citation on regarding a temperature drift correction has been added after Eq. 7.

Equation 14 has been corrected.

The comment concerning future satellite missions in section 4.5 is elaborated on.

- Page 1, line 11: Please correct: "This manuscript reports on the data available from and airborne gravity campaign carried out in summer 2023." into "This manuscript reports on the data available from **an** airborne gravity campaign carried out in summer 2023" Done
- Page 1, line 14: I suggest over-emphasising this statement somewhat: "Unlike classical technologies that can only measure the variation of gravity from an aircraft..." → "Unlike classical technologies that can only measure the relative variation of gravity from an aircraft..."
  - Done
- Page 3, figure 1: I recommend to indicate the Snæfellsjökull volcano in the map as well Done
- Page 11, line 181 ff: The sentence "... the iMAR strapdown sensor are corrupted by the presence of systematic errors in the gravity sensor" sounds a bit like that the strapdown sensor is faulty. I recommend such a formulation: "Following the processing described in the previous section, the relative gravity disturbance estimates derived from the iMAR strapdown sensor need to be corrected for the therein contained systematic errors."

This has been corrected.

 Page 11, table 2: To avoid misunderstandings, please change "Height" into "Ellipsoidic height" in the header. This has been corrected. • Page 12, figure 6: Could you specify over which location the plotted tracks were taken? This has been added to the caption.